

# 1  Quantification of atmospheric nucleation and growth process
# 2  as a single source of aerosol particles in a city

Imre Salma, Veronika Varga, Zoltán Németh
Institute of Chemistry, Eötvös University, H-1518 Budapest, P.O. Box 32, Hungary
*Correspondence to*: Imre Salma (salma@chem.elte.hu)
**Abstract.** Effects of new aerosol particle formation (NPF) and particle diameter growth process as a single source on
atmospheric particle number concentrations were evaluated and quantified on the basis of experimental data sets obtained from
particle number size distribution measurements in the city centre and near-city background of Budapest for 5 years. Nucleation
strength factors separately for nucleation days ($NSF_{\text{nucl days}}$) and for all days ($NSF_{\text{all days}}$) were derived for seasons and full years.
The former characteristics represents the concentration increment of ultrafine (UF) particle numbers with respect to
background concentration due solely to nucleation specifically on nucleation days. The latter factor expresses the contribution
of nucleation process to the background particle number concentrations in general, thus on a longer time interval such as season
or year. The nucleation source had the largest effect on particle concentrations around noon and early afternoon as expected.
During this time interval, it became the major source of particles in the near-city background. Nucleation increased the daily
mean particle number concentrations on nucleation days by mean factors of 2.3 and 1.58 in the near-city background and city
centre, respectively. Its effect was the largest in winter, which was explained with the substantially lower background
concentration levels on nucleation days than that on non-nucleation days. On an annual time scale, 37% of the UF particles
were generated by nucleation in the near-city background, while NPF produced 13% of UF particles in the city centre. The
differences among the annual mean values, and among the corresponding seasonal mean values were likely caused by the
variability in controlling factors from year to year. The values obtained represent lower limits of contributions. The shares
determined imply that NPF is a non-negligible or substantial source of particles in near-city background environments and
even in city centres, where the vehicular road emissions usually prevail. Atmospheric residence time of nucleation-mode
particles was assessed by decay curve analysis of $N_{6-25}$ concentrations in time, and a mean of 2:30 was obtained. The present
study suggests that the health-related consequences of atmospheric NPF and growth process in cities should also be considered
in addition to its urban climate implications.

## 26  1 Introduction

Large-scale modelling studies suggest that new aerosol particle formation (NPF) and consecutive particle diameter growth
process in the atmosphere (Kulmala et al., 2004, 2013) is the dominant source of particle number concentrations on global
scale (Spracklen et al., 2006; Reddington et al., 2011; Makkonen et al., 2012; Yu et al., 2015). In addition, up to approximately
50% of all cloud condensation nuclei (CCN) can originate from NPF and growth (Spracklen et al., 2008; Merikanto et al.,
2009), which relates the process to the climate system, and indicates its overall importance (Kerminen et al., 2012; Carslaw et
al., 2013; Shen et al., 2017). New particle formation has also been proved to be common in large cities (Nieminen et al., 2017).
Urban NPF can interact with and can be influenced by regional nucleation events at least under some geographic conditions,
and can become part of a phenomenon with a much larger horizontal extension than the city (Salma et al., 2016b). At the same
time, particle number concentrations in cities are strongly affected by high-temperature emission sources from different sectors
(Paasonen et al. 2016) such as household and residential heating (e.g. Butt et al., 2016), industrial processes and power
production (e.g. Xiao et al., 2015), and vehicular road traffic (e.g. Morawska et al., 2008). Their diurnal variability often show
daily time-activity pattern of inhabitants (Dall'Osto et al., 2013). Relative contributions of primary and secondary particle
sources – particularly in cities – change substantially in time and space (Pikridas et al., 2015; Posner and Pandis, 2015). Several



methods were proposed to distinguish the major production types of particles (e.g. Shi et al., 1999; Alam et al., 2003; Rodrígues
and Cuevas, 2007; Qian et al., 2007; Park et al., 2008; Costabile et al., 2009; Brines et al., 2015). The share of NPF as a single
source of ambient particle number concentrations specifically in cities remained, however, largely unknown. Despite the fact
that there is often a spatial coincidence between the poorer air quality and population density (Samoli et al., 2016). Moreover,
approximately 70–80% of total particles in cities belong to the ultrafine (UF) size range (with an equivalent diameter <100
nm; Putaud et al., 2010), and their inhalation can represent an excess health risk relative to coarse or fine particles with the
same or similar chemical composition (Oberdörster et al., 2005; Braakhuis et al., 2014). An estimate on the relative contribution
of primary and secondary formation processes is also required for efficient action plans to improve the air quality in cities. It
is worth noting that indirect climate effects (due to CCN) become important for particles with a diameter >50–100 nm, while
the excess health effects are linked with diameters <100 nm.

Nucleation strength factor (NSF) was introduced to assess the contribution of NPF to UF particle number concentrations
relative to the background with respect to all other sources (Salma et al., 2014). The results derived from this approach
correspond to the mode-segregated secondary particle load. By now, atmospheric concentration data sets are available for
multiple years to study the applicability and behaviour of NSF in detail. The major advantage of this quantification is that it
only requires experimental data that can be readily derived from ordinary NPF (size distributions) measurements. The main
objectives of this paper are to quantify and discuss the contribution of NPF events to ambient particle number concentrations
in near-city and central urban environments of a Central European city considering five-year long data sets, to investigate and
explain the details of the NSF, and to interpret the consequences achieved for the urban air and air quality.
**2 Methods**
**2.1 Experimental**
The measurements were performed in Budapest, Hungary. Its population is approximately 2.5 million in the metropolitan area.
The major pollution sources in terms of particle number include vehicular road traffic, residential heating and household
burning activities. Contributions of passenger cars and buses to the vehicle fleet registered in Budapest and Pest County are
87% and 0.46%, respectively (OKJ, 2015). Diesel-powered vehicles shared 19% and 97% of the national passenger car and
bus fleets, respectively. Wintertime median concentrations of particulate matter (PM) mass, elemental carbon (EC) and organic
carbon (OC) in the $PM_{2.5}$ size fraction were 25, 0.97 and 4.9 μg m$^{-3}$, respectively in the related time interval (Salma et al.,
2017). The mean contributions of EC and organic matter (OM, with an OM/OC mass conversion factor of 1.6) to the $PM_{2.5}$
mass and standard deviations (SDs) were 4.8±2.1% and 37±10%, respectively, while the contribution of $(NH_4)_2SO_4$ and
$NH_4NO_3$ derived from an earlier study in spring were 24% and 3%, respectively. The contributions of EC and OC from fossil
fuel combustion to the TC were 11.0% and 25%, respectively, and EC and OC from biomass burning were responsible for
5.8% and 34%, respectively of the TC, while the OC from biogenic sources made up 24% of the TC.

Two urban sites were involved in the study. Most measurements were performed at the Budapest platform for Aerosol Research
and Training (BpART) facility (N 47° 28' 29.9", E 19° 3' 44.6", 115 m above mean see level (a.s.l.) of the Eötvös University
(Salma et al., 2016a). The sampling inlets were set up at heights between 12 and 13 m above the street level. The location
represents a well-mixed, average atmospheric environment for the city centre. The other location was situated at the NW border
of Budapest in a wooded area of the Konkoly Astronomical Observatory of the Hungarian Academy of Sciences (N 47° 30'
00.0", E 18° 57' 46.8", 478 m a.s.l.). It represents the air masses entering the city since the prevailing wind direction in the
Budapest area is NW. The experimental data obtained for five full-year long time intervals, i.e. from 03–11–2008 to 02–11–
2009, from 19–01–2012 to 18–01–2013, from 13–11–2013 to 12–11–2014, from 13–11–2014 to 12–11–2015 and from 13–





11–2015 to 12–11–2016 were considered in the present study. Local time (UTC+1 and daylight saving time, UTC+2) was
chosen as the time scale because the daily routine activities of inhabitants in cities were primarily considered.

The key measuring instrument was a flow-switching type differential mobility particle sizer (DMPS; Salma et al., 2011). Its
main components include a Ni-60 radioactive bipolar charger, a Nafion semi-permeable membrane dryer, a 28-cm long
Vienna-type differential mobility analyser and a butanol-based condensation particle counter (TSI, model 3775). The system
operates in an electrical mobility diameter range from 6 to 1000 nm in the dry state of particles (with a RH<30%) in 30 channels
with a time resolution of approximately 8 or 10 min at two sets of flows. The sample flow rate is 2.0 L min$^{-1}$ in high-flow
mode, and 0.31 L min$^{-1}$ in low-flow mode with sheath air flow rates 10 times larger than for the sample flows. The DMPS
measurements were performed according to the recommendations of the international technical standard (Wiedensohler et al.,
2012). The DMPS data for the 1-year long time intervals in 2008–2009, 2012–2013, 2013–2014, 2014–2015 and 2015–2016
were available in 95%, 95%, 99%, 95% and 73% of the total number of days, respectively. Meteorological data were recorded
by an on-site meteorological station (Salma et al., 2016a). Standardised meteorological measurements of air temperature ($T$),
relative humidity (RH), wind speed (WS) and wind direction (WD) were recorded with a time resolution of 10 min. The
coverage of the meteorological data was >80% in each year.
**2.2 Data treatment**
The overall treatment of the measured DMPS data was performed according to the procedure protocol by Kulmala et al. (2012).
The inverted DMPS data were utilised to generate particle number size distribution surface plots showing jointly the variation
in particle diameter and particle number concentration density in time. Identification and classification of NPF and growth
processes was accomplished from the surface plots by using the algorithm similar to that of Dal Maso et al. (2005) on a day-
to-day basis into the following main classes: NPF event days, non-event days, days with undefined character, and days with
missing data (for more than 4 h in the midday). Frequency of events was determined as the ratio of the number of event days
to the total number of relevant (i.e. all–missing) days. Particle number concentrations in the diameter ranges from 6 to 1000
nm ($N_{6-1000}$), from 6 to 100 nm ($N_{6-100}$), from 6 to 25 nm ($N_{6-25}$) and from 100 to 1000 nm ($N_{100-1000}$) were calculated from the
DMPS data. The major portion of the $N_{6-100}$ concentration (i.e. the Aitken mode) is essentially related to local source processes
due to their limited atmospheric residence time (typically <$10^1$ h), while the $N_{100-1000}$ (concentration mainly in the accumulation
mode) expresses larger (background) spatial and time scales because of much longer residence times (up to $10^1$ d; Salma et
al., 2011). To derive mean diurnal variability, the exact recording times belonging to individual concentrations were rounded
off to 5 min (in case of the time resolution of ca. 8 min) or 10 min (in case of the time resolution of ca. 10 min) time scale. The
concentrations and further properties derived from them (see later) were averaged by the time of day separately for nucleation
and non-nucleation days, and in a year. Finally, the averaging was also performed separately for different seasons, hence for
spring (March-May), summer (June-August), autumn (September-November) and winter (December-February).

Two types of NSF (Salma et al., 2014) were derived in the present study by considering different conditions. The quantity:
$$\mathrm{NSF}_{\mathrm{nucldays}} = \frac{\left(N_{6-100} \big/ N_{100-1000}\right)_{\mathrm{nucleation\,days}}}{\left(N_{6-100} \big/ N_{100-1000}\right)_{\mathrm{non\text{-}nucleation\,days}}} \qquad (1)$$

considers the $N_{6-100}/N_{100-1000}$ concentration ratios for nucleation days only. The numerator expresses the increase in $N_{6-100}$
concentration relative to the background concentration $N_{100-1000}$ caused by all source sectors. The denominator represents the
same property due to all sources except for NPF. Hence, the $\mathrm{NSF}_{\mathrm{nucl\,days}}$ accounts for the increment in background particle
concentration on nucleation days exclusively caused by NPF. It was implicitly assumed that the major emission and formation





processes of UF particles except for NPF are uniformly present on both nucleation and non-nucleation days. It seems to be a
reasonable condition for time intervals of several months, although the number of nucleation days during a time interval
actually plays a more determining role than the length of the interval. Winter, when the occurrence frequency shows the
minimum (see Table 1), appears to be the most restrictive season. The effect of the non-uniformly present sources is indicated
by unusually larger scatter in the diurnal data points (see Sect. 3.2). It was also presumed that the production of particles larger
than 100 nm was much smaller than the concentration of UF particles. This is ordinarily realised in cities, and can be justified
from the contributions of UF particles to the total particle number (Putaud et al., 2010; Németh et al., 2017).

The other type of NSF was calculated for all days in the numerator, thus:
$$\mathrm{NSF}_{\mathrm{all\,days}} = \frac{\left( N_{6-100} \middle/ N_{100-1000} \right)_{\mathrm{all\,days}}}{\left( N_{6-100} \middle/ N_{100-1000} \right)_{\mathrm{non\text{-}nucleation\,days}}}$$
(2)

It expresses the overall contribution of NPF to particle numbers of background concentration on longer time span or in general.
Since there are usually more non-nucleation days than nucleation days in a time interval of month or more, the assumptions
for $\mathrm{NSF}_{\mathrm{all\,days}}$ are met easier than for $\mathrm{NSF}_{\mathrm{nucl\,days}}$. The $\mathrm{NSF}_{\mathrm{nucl\,days}}$ characterises an ordinary nucleation day within e.g. a season,
while the $\mathrm{NSF}_{\mathrm{all\,days}}$ quantifies the overall effect of NPF and growth on the atmospheric concentrations on a typical or average
day over e.g. a season or year. If 1) NSF≈1 then the relative contribution of nucleation to particle number concentrations with
respect to other sources is negligible, 2) 1<NSF<2 then its relative contribution as a single source is considerable, and 3)
NSF>2 then the contribution of nucleation itself to particle number concentrations is larger than of any other source sectors
together. Since the major phase of NPF and growth process takes place in most cases in one day, it is advantageous to express
NSFs as daily mean values. The data for the undefined days were not taken into account for the present evaluation.
**3 Results and discussion**
Number of nucleation days for different seasons in each measurement year are summarised in Table 1. It is seen that the NPF
frequency has an obvious seasonal variability. This can be obtained from its monthly dependency which exhibits an absolute
and local minimum in January and August, respectively, and an absolute and local maximum in March or April, and September,
respectively (Salma et al., 2016b). The seasonal variation of the nucleation frequency fits into the second group of the
measurement sites reported by Manninen et al. (2010).
**Table 1.** Number of nucleation days for seasons in the near-city background (in 2012–2013) and in the city centre (in 2008–2009, 2013–
2014, 2014–2015 and 2015–2016) during 1-year long time intervals.

| Environment | Time interval | Spring | Summer | Autumn | Winter |
|-------------|---------------|--------|--------|--------|--------|
| Background  | 2012–2013     | 35     | 20     | 24     | 17     |
| Centre      | 2008–2009     | 34     | 21     | 22     | 6      |
| Centre      | 2013–2014     | 28     | 20     | 13     | 11     |
| Centre      | 2014–2015     | 41     | 19     | 14     | 7      |
| Centre      | 2015–2016     | 15     | 9*     | 5*     | 6      |

* Low data coverage. See Sect. 2.1.





### 3.1 Seasonal atmospheric concentrations
Particle number concentrations in the related size fractions for different seasons in each year are summarised in Table 2 for an
overview.
**Table 2.** Median atmospheric concentration of particles with a diameter from 6 to 100 nm ($N_{6-100}$) and from 100 to 1000 nm ($N_{100-1000}$) in
units of $10^3$ cm$^{-3}$ separately on nucleation (Nucl) days and non-nucleation (Nonucl) days for seasons in the near-city background (in 2012–
2013) and in the city centre (in 2008–2009, 2013–2014, 2014–2015 and 2015–2016) during 1-year long time intervals. Mean ratios of median
concentrations on nucleation days to that on non-nucleation days with standard deviations (SDs) for the size fraction are also indicated.

| Urban environment | | | Near-city backgr. | | City centre | | | | |
|---|---|---|---|---|---|---|---|---|---|
| Season | Size fraction | Day type | 2012–2013 | Ratio | 2008–2009 | 2013–2014 | 2014–2015 | 2015–2016 | Ratio ±SD |
| Spring | $N_{6-100}$ | Nucl | 4.8 | 1.72 | 11.2 | 9.7 | 10.0 | 8.6 | 1.37 |
| | $N_{6-100}$ | Nonucl | 2.8 | | 8.9 | 7.2 | 7.1 | 5.9 | ±0.09 |
| | $N_{100-1000}$ | Nucl | 1.56 | 1.03 | 2.0 | 2.5 | 2.6 | 1.56 | 0.99 |
| | $N_{100-1000}$ | Nonucl | 1.51 | | 2.2 | 2.7 | 2.5 | 1.5 | ±0.07 |
| Summer | $N_{6-100}$ | Nucl | 4.0 | 1.37 | 10.3 | 8.0 | 8.6 | 6.9 | 1.17 |
| | $N_{6-100}$ | Nonucl | 2.9 | | 8.9 | 7.5 | 6.5 | 6.2 | ±0.11 |
| | $N_{100-1000}$ | Nucl | 1.27 | 0.89 | 1.36 | 2.0 | 2.5 | 1.40 | 0.92 |
| | $N_{100-1000}$ | Nonucl | 1.42 | | 1.72 | 2.4 | 2.4 | 1.37 | ±0.13 |
| Autumn | $N_{6-100}$ | Nucl | 4.3 | 1.29 | 14.0 | 11.9 | 12.6 | 5.2 | 1.41 |
| | $N_{6-100}$ | Nonucl | 3.3 | | 10.4 | 8.5 | 8.4 | 5.1 | ±0.08 |
| | $N_{100-1000}$ | Nucl | 1.67 | 0.74 | 2.0 | 3.4 | 2.7 | 1.6 | 0.87 |
| | $N_{100-1000}$ | Nonucl | 2.3 | | 2.4 | 3.9 | 3.3 | 1.7 | ±0.06 |
| Winter | $N_{6-100}$ | Nucl | 3.9 | 1.10 | 6.9 | 10.5 | 5.6 | 7.7 | 0.87 |
| | $N_{6-100}$ | Nonucl | 3.6 | | 12.5 | 9.2 | 7.8 | 7.4 | ±0.28 |
| | $N_{100-1000}$ | Nucl | 1.12 | 0.38 | 1.02 | 3.7 | 1.65 | 1.4 | 0.54 |
| | $N_{100-1000}$ | Nonucl | 2.9 | | 3.0 | 4.5 | 3.8 | 2.7 | ±0.22 |


Data coverage for summer and autumn in 2015–2016 were low, and therefore, the corresponding concentration ratios were
excluded from the averaging for the mean ratios. It can be seen that the $N_{6-100}$ were ordinarily larger on nucleation days than
on non-nucleation days. This is likely a direct effect of nucleation. At the same time, the $N_{100-1000}$ usually showed a constant
level within approximately 10% except for winters and some autumns. The background concentrations on nucleation days
were smaller (by factors of 0.4–0.5) than for non-nucleation days particularly in winter. The differences are further discussed
and explained in Sect. 3.2.
### 3.2 Concentration increment on nucleation days
Diurnal variability of the concentration increment due to NPF on nucleation days (i.e. NSF$_{\text{nucl days}}$) for the city centre and near-
city background separately for different seasons are shown in Fig. 1 as representative examples. The curves exhibited a single
peak around noon with a longer tail on the decreasing side. The exact location of the peak is also influenced by setting the
local daylight saving time in spring and autumn. The baseline of some peaks from 0:00 to 7:00 deviated systematically and
substantially from unity although no nocturnal nucleation has been observed in Budapest. The mean values of this baseline in
the city centre for spring, summer, autumn and winter (Fig. 1 upper panel) were 1.02, 1.15, 1.15 and 1.55, respectively, while
they were 1.11, 1.18, 1.31 and 1.72, respectively in the near-city background (Fig. 1 lower panel). This elevated line can be
explained by the fact that particle growth process could be traced till the late morning of the next day in several occasions,
thus the NPF influenced the $N_{6-100}$ concentrations over the next morning. This affected the baseline if a non-nucleation day
followed a nucleation day, and particularly, in the seasons when NPF events occur well separated from each other in time,
which is typical for winter. The elevated line is a real effect of NPF, and its consideration in the averaging is justified.




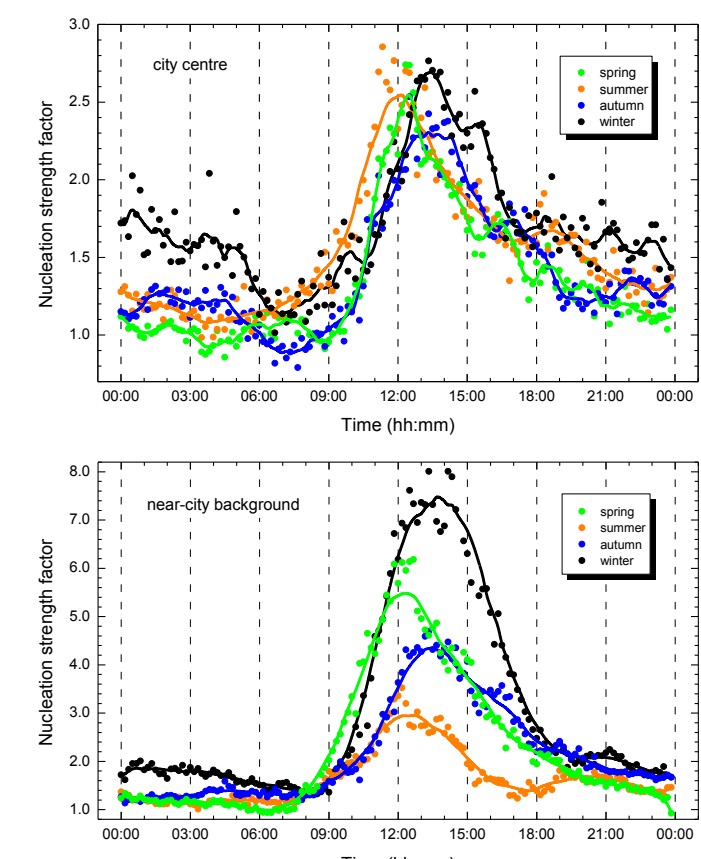

**Figure 1.** Diurnal variability of concentration increment due to NPF on nucleation days (nucleation strength factor $NSF_{nucl\ days}$) for the city
centre in 2008–2009 (upper panel) and in the near-city background in 2012–2013 (lower panel) separately for seasons. The solid lines
represent 1-h smoothing to the data.

It was also observed in all years that the concentration increment on nucleation days due to NPF (i.e. $NSF_{nucl\ days}$) was the
largest for winter. This evidently showed up for the near-city background. It was followed by the other seasons which had
similar importance to each other in the city centre, or which were ordered as spring, autumn and summer in the near-city
background. To investigate these findings more closely, diurnal variability of the related particle number concentrations were
derived and evaluated. Diurnal variability of the concentrations for summer and winter are shown in Figs. 2 and 3, respectively
for the city centre and near-city background. The dependencies in the city centre for the spring, other summer and autumn
seasons are similar to Fig. 2 upper panel, while the corresponding seasonal curves for the near-city background resemble Fig.
2 lower panel. The diurnal patterns represented by Fig. 2 are coherent with the previous ideas on the NPF and growth events
in the Budapest area (Salma et al., 2014, 2016b; Németh et al., 2017), and they also confirm the basic assumptions of the NSF
definition on the source intensities at both location types. The comparison of the $N_{6-100}$ curves for nucleation days and non-
nucleation days already emphasizes the importance of NPF, and indicates that the phenomenon has larger relative effect in the
near-city background than in the central urban parts as it is expected. It is worth realising that the particle number
concentrations for the background aerosol ($N_{100-1000}$) appear close to each other within a relative uncertainty of 10–20%, which
implies that the background concentrations affect the NPF occurrence and formation rate in a limited manner in these seasons.



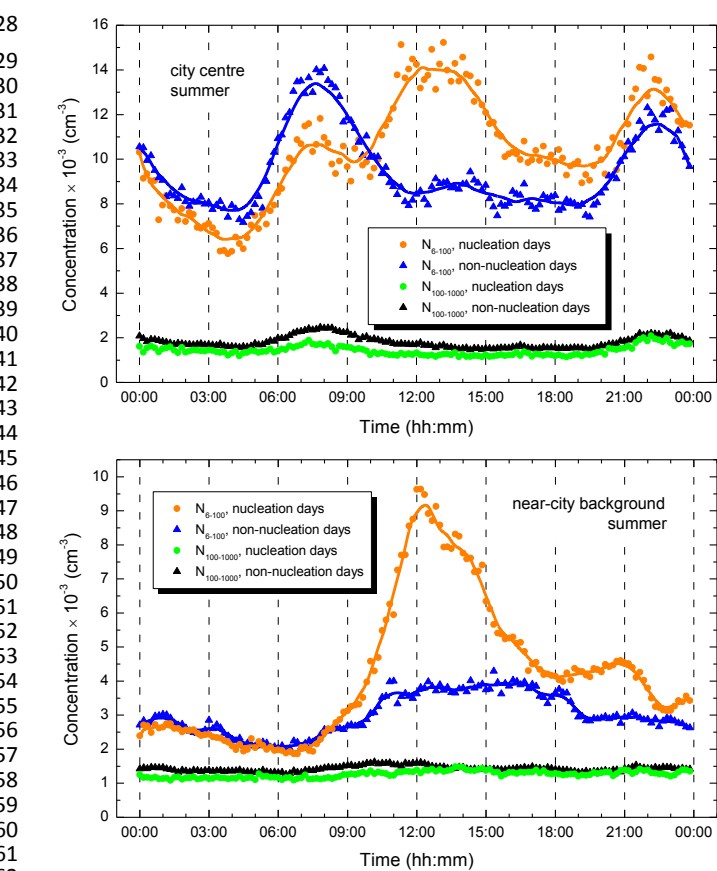

**Figure 2.** Diurnal variability of particle number concentrations in summer for the diameter ranges from 6 to 100 nm ($N_{6-100}$) and from 100 to 1000 nm ($N_{100-1000}$) in the city centre in 2008–2009 (upper panel) and in the near-city background in 2012–2013 (lower panel) separately for nucleation days and non-nucleation days. The solid lines represent 1-h smoothing to the data.

For winter, it is seen, however, that the background concentrations were substantially different for the nucleation and non-nucleation days (Fig. 3). The mean non-nucleation/nucleation $N_{100-1000}$ ratios for the city centre (in 2008–2009) and near-city background were 2.8 and 2.3, respectively. This implies that NPF events preferably took place on those days when the particle number concentrations were generally smaller. It is understandable if we consider that the basic preconditions of NPF events are realised by competing source and sink for condensing vapours. The source strength in winter has a decreased tendency due to lower solar radiation intensities and less (biogenic) precursor gases in the air (Salma et al., 2011). Nevertheless, nucleation can occur at these small source terms if the (condensation and scavenging) sink - which is related to the concentration of pre-existing particles - is even smaller. This explains the differences in the background concentrations on nucleation and non-nucleation days. Larger concentration increments (higher $NSF_{nucl\ days}$, Fig. 3) for winter were simply caused by systematically smaller background concentrations on nucleation days. In addition, the fluctuation in the $N_{100-1000}$ for nucleation days in winter of the other years was sometimes larger than that shown in Fig. 3. This observation raises the question which is the smallest number of NPF events in a time interval (e.g. season), which can be considered to be sufficient for obtaining representative mean diurnal concentration data for calculating the NSF. A few NPF events during winters (see Table 1) might not be fully satisfactory for this purpose.



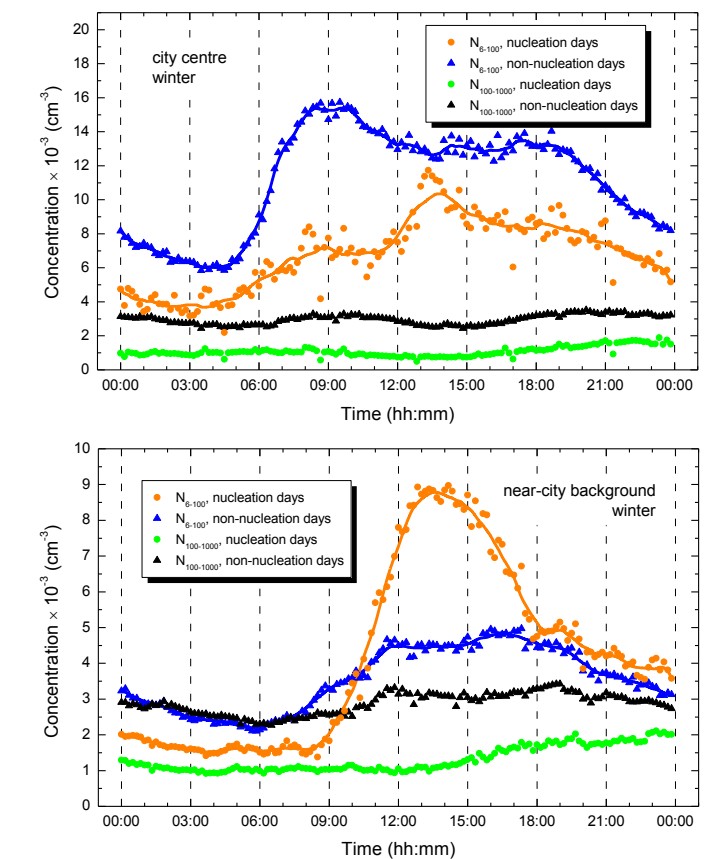

**Figure 3.** Diurnal variability of particle number concentrations in winter for the diameter ranges from 6 to 100 nm ($N_{6-100}$) and from 100 to 1000 nm ($N_{100-1000}$) in the city centre in 2008–2009 (upper panel) and in the near-city background in 2012–2013 (lower panel) separately for nucleation days and non-nucleation days. The solid lines represent 1-h smoothing to the data.

Seasonal and annual mean concentration increments of background concentration on nucleation days (hence the daily mean NSF$_{\text{nucl days}}$ values) for different years are summarised in Table 3. Nucleation as a single source increased the daily particle number concentrations by factors of 2.3 and 1.58 in the near-city background and city centre, respectively on an annual time scale. The differences among the annual mean values, and among the corresponding seasonal mean values were likely caused by the variability of controlling parameters from a year to year. As far as the seasonal variability is concerned, it is noted that the formation rate for particles with a diameter of 6 nm ($J_6$) showed only rather modest seasonal dependency in Budapest (Salma et al., 2011), which also contributed to similar mean increments for spring, summer and autumn.

**Table 3.** Seasonal and annual mean increments of background concentration due to nucleation on nucleation days (nucleation strength factor NSF$_{\text{nucl days}}$) in the near-city background (in 2012–2013) and in the city centre (in 2008–2009, 2013–2014, 2014–2015 and 2015–2016) during 1-year long time intervals.

| Environment | Time interval | Spring | Summer | Autumn | Winter | Year |
|---|---|---|---|---|---|---|
| Background | 2012–2013 | 2.3 | 1.66 | 2.2 | 3.0 | 2.3 |
| Centre | 2008–2009 | 1.36 | 1.55 | 1.42 | 1.71 | 1.49 |
| Centre | 2013–2014 | 1.31 | 1.33 | 1.36 | 1.56 | 1.44 |
| Centre | 2014–2015 | 1.50 | 1.34 | 1.83 | 2.8 | 1.73 |
| Centre | 2015–2016 | 1.54 | 1.26 | 1.46 | 2.4 | 1.64 |



### 3.3 Contribution of nucleation to particle number concentrations

Diurnal variability of NSF$_{all\ days}$ is shown in Fig. 4. The curves exhibited a single peak with a maximum around noon and a longer tail in the early afternoon. The maximum values in the city centre represented concentration contributions from 30% to 60% due to nucleation for a limited time interval. The curve for the near-city background was the largest, as expected, and it even exceeded the value of 2 around noon for approximately 3 h. This all means that nucleation has an important contribution to UF particles during the midday in the city centre, while it even becomes the dominant source of particles directly after midday in the near-city background.

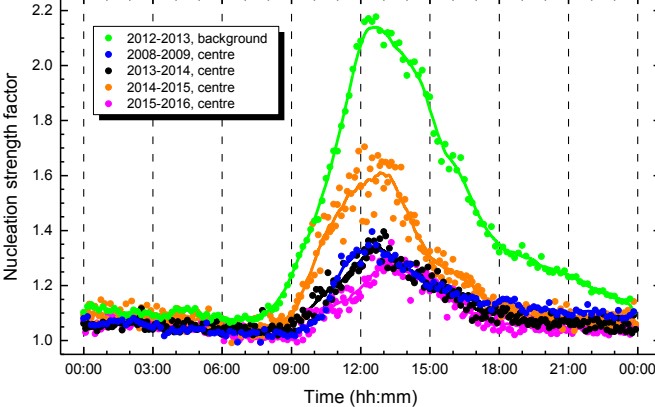

**Figure 4.** Diurnal variability of NPF contribution to particle number concentrations (nucleation strength factor NSF$_{all\ days}$) in the near-city background (in 2012–2013) and in the city centre (in 2008–2009, 2013–2014, 2014–2015 and 2015–2016) during 1-year long time intervals. The solid lines represent 1-h smoothing to the data.

The importance of nucleation was also demonstrated for different seasons and years by the mean NSF$_{all\ days}$ values which are summarised in Table 4. In general, 37% of UF particles (more precisely of particle number increment) were produced by nucleation as a single source in the near-city background. In the city centre, it generated 13% of UF particles. These values can be considered as lower limits since a considerable part of the background particles is also produced by NPF. The differences among the annual mean values, and among the corresponding seasonal mean values were likely caused by the year-to-year variability similarly to the concentration increments on nucleation days (NSF$_{nucl\ days}$). It is informative to compare the contribution values to the global share of various source sectors in primary UF particle number emission to have an idea on the relative extent of our results. It is stressed that our and the literature data are related to very different types of particles, nevertheless, some analogy can be found in the relative importance of the sources and source processes. Road transport, power production and residential combustion are the first three largest contributors to primary UF particles with the shares of 40%, 20% and 17%, respectively (Paasonen et al., 2016). The actual contributions can vary in different parts of the world and with economic development.

The effects of particles generated by NPF and growth in increased concentrations on human health and the environment are also influenced by the time interval for which the particles remain in the air. Nucleation-mode particles are primarily removed by coagulation with larger particles, agglomeration, diffusion and turbulent losses, scavenging and various aging processes. We showed in Sect. 3.2 and it can be also proved directly from the measured data that it is the NPF events that usually produce the $N_{6-25}$ concentrations in an overwhelming extent (the $N_{6-25}$ is increased by 1–2 orders of magnitude) in a relatively short time interval. The continual concentration decrease in time for several hours after the event can be utilised to assess the general atmospheric residence time of nucleation-mode particles. The decrease of this concentration after a nucleation burst could be





approximated by an exponential function (first order kinetics). By a decay curve analysis of 15 selected NPF and particle
diameter growth cases, the residence times were estimated from the slope of the concentrations $N_{6-25}$ in a natural logarithmic
scale versus time. The residence time of nucleated particles varied from 1:30 to 4:15 with a mean and SD of 2:30±1:00. This
suggests that the nucleation-mode particles (or in other words the nucleated particles in their very small sizes, or atmospheric
nanoparticles) likely have limited health effects due to their relatively short existence in the air.
**Table 4.** Seasonal and annual mean contributions of nucleation to background concentration (nucleation strength factor $NSF_{all\ days}$) in the
near-city background (in 2012–2013) and in the city centre (in 2008–2009, 2013–2014, 2014–2015 and 2015–2016) during 1-year long time
intervals.

| Environment | Time interval | Spring | Summer | Autumn | Winter | Year |
|---|---|---|---|---|---|---|
| Background | 2012–2013 | 1.51 | 1.18 | 1.31 | 1.40 | 1.37 |
| Centre | 2008–2009 | 1.12 | 1.15 | 1.10 | 1.05 | 1.12 |
| Centre | 2013–2014 | 1.11 | 1.07 | 1.08 | 1.11 | 1.11 |
| Centre | 2014–2015 | 1.22 | 1.08 | 1.14 | 1.16 | 1.19 |
| Centre | 2015–2016 | 1.09 | 1.06 | 1.03 | 1.09 | 1.09 |

**4 Conclusions**
We showed in the present study that NPF and particle diameter growth process as a single source represents a considerable
contribution to UF particles in a Central European city with respect to all other emission sources including vehicular road
traffic. Nucleation was a major process that produced UF particles at noon and in the early afternoon, and its relative
contribution was comparable to other production sources during this time period even in the city centre. Relative importance
of nucleation as a source of particles decreased with anthropogenic influence. The NSFs were defined by utilising $N_{6-100}$ and
$N_{100-1000}$ concentrations. There are several sensible and practical reasons for selecting these specific size fractions although
other dividing values are also imaginable. The quantifications in the present study are, therefore, subjected to certain inherent
uncertainty. Considering that the modes of the particle number size distribution are usually shifted to smaller diameters in
cities with respect to rural or remote areas, it seems realistic that these size fractions represent well the particles of urban (local)
origin and the aged particles (which characterize larger spatial or urban background area), respectively. The study also suggests
that particles from NPF events in cities are relevant not only for their effects on urban climate but because of their health risk
to inhabitants. At the same time, it should also be mentioned that ambient atmospheric aerosol which ordinary persists in the
air of cities contains particles in the largest abundance with a diameter between approximately 25 and 150 nm. Smaller particles
are thermodynamically not stable, and most of them are removed from the air during relatively short time intervals. The
exposures to freshly nucleated particles ($d$<25 nm) or ambient nanoparticles ($d$<10 nm) are usually limited to several hours
after the onset of the NPF.

Regulations of aerosol emissions and atmospheric concentrations are usually based on PM mass. The changes or reductions in
anthropogenic aerosol load are ordinary assessed by assuming similar relative tendencies in particle mass and particle number
concentrations. This generalisation may yield to tentative conclusions. According to current legislation scenarios, particle
number emissions are expected to decrease in most part of the world by 2030 mainly due to spreading the diesel particulate
filters (DPF) in cars and due to diesel fuels with ultralow sulphur content. In effect, this may imply that the relative share of
NPF in the particle number production is expected to increase above the levels estimated in the present study. By demonstrating
the relevance of NPF as an important single source of UF particles, we also raise the question of an international enhanced
particle mass and particle number inventory with precursor gas data that potentially includes the NPF and growth process as a
separate sector among the source types.





## 5 Data availability

The relevant observational data used in this paper are available on request from the corresponding author or at the website of

the Budapest platform for Aerosol Research and Training (http://salma.web.elte.hu/BpArt).

*Acknowledgement.* Financial supports by the National Research, Development and Innovation Office of Hungary (contract

K116788) is appreciated.

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
