# Peer review of "as a single source of aerosol particles in a city"

_Atmospheric Chemistry and Physics, 2017_

## Referee Comment (RC1) · Anonymous Referee #1 · 24 Oct 2017

This manuscript evaluates the relative importance of new particle formation (NPF) events as a single source in terms of its contribution to atmospheric particle number concentrations. This evaluation is based atmospheric particle number size distribution measurements in the city center and near-city background of Budapest for 5 years, using a Nucleation strength factor, NSF. The conclusions of this manuscript are interesting but not surprising. I would like to recommend publication of this manuscript in Atmospheric Chemistry and Physics after the following concerns are addressed.

1. Two NSF factor have been used in the manuscript. The mixed usage of the two factors always confused me. I would like to suggest the authors to clearly state which

factor they are referring to whenever possible, e.g., NSFs in Line 134-136. Also, is NSFnucl days larger than NSFall days? Looks to me that (N6-100/N100-1000)nul days is larger than (N6-100/N100-1000)all days by definition, isn't it? I would like to see more discussion on the relationship between the two factors.

2. In table 2, wouldn't I get (N6-100/N100-1000)nul days from the numbers listed? For example, (N6-100/N100-1000)nul days can be obtained (1.72/1.03 = ((N6-100)Nucl/(N6-100)Non-nucl)/ ((N100-1000)Nucl/(N100-1000)Non-nucl) = ((N6-100)Nucl/(N100-1000)Nucl)/((N6-100)Non-nucl/(N100-1000)Non-nucl) = (N6-100/N100-1000)nul days. However, my number is different from those listed in Table 3. What is the problem?

3. (Line 156-158), put "Data coverage for summer and autumn . . . for the mean ratios" as a footnote of Table 2.

4. (Line 159-160) the lower background particle concentration on nucleation days in winter came from real measurements, right?

5. (Line 173) rephrase "its consideration in the averaging is justified".

6. Clearly mark the vertical coordinates (NSFnucl days or NSFall days) in Figures 1 & 4, and also include NSFnucl days or NSFall days in the body instead of the title of Tables 3 & 4.

7. In table 2, what caused the behavior of N6-100 on NPF days? Especially, the peak at night. Also, please expand to discuss why N6-100 is significantly larger between 6-9 am on non-nucleation days? Does this mean that nucleation was hindered by the high concentrations of preexisting particles?

8. (Line 272) lower solar radiation in winter is understandable, but less biogenic precursor gases are not justified. Is there any evidence that NPF in Budapest requires biogenic vapors?

9. (Line 277-279) rephrase "this observation raises the question. . .for calculating the

NSF".

10. Regarding the health effects (Line 385-387), I would like to suggest the authors to be more conservative. The relative short lifetime is one aspect, but the toxicity per particle is another aspect. The authors just can't evaluate the health effects of nanoparticles generated by NPF.
* * *

---

## Referee Comment (RC2) · Anonymous Referee #2 · 31 Oct 2017

The authors present an analysis on the impact of regional new particle formation (NPF) events on the aerosol particle number concentrations in urban and near-urban environments. They aim to quantify this impact by applying different versions of nucleation strength factors (NSFs), which are determined to describe the relative impact of NPF events on the particle concentrations. The goal of the study is of importance for considerations of improving urban air quality. However, I find that the main parameter applied in this study, NSF, might not be suitable for drawing accurate conclusions, and should at least be investigated and explained in more detail. Based on my main comment below, I cannot recommend the publication of this manuscript without further analysis on the behavior of the parameter NSF.

[Figure]

Major comment

The actual meaning of the NSF is not clear, due to the normalization of nucleation + Aitken mode concentration with the accumulation mode (why not e.g. with the mean N6-100 from 6 to 9 am on the same morning?). The applied normalization may cause unintended signals: for example, if we consider two days during which the N6-100 is exactly similar, but on the latter (called here a nucleation day just to show the point) N100-1000 is lower than on the first by a factor of 1.5, the NSF would by 1.5. This would not only be a false signal but also to the wrong direction: during lower sink conditions the equal source should lead to higher concentrations, and if equal N6-100 was observed, the source should be weaker and thus NSF smaller than 1. This does not necessarily mean that the applied definition of NSF would not make sense, but it's behavior with the applied data set should be analyzed and its meaning explained much more in detail.

Minor comments

On the terminology:

The word nucleation is used in the manuscript for regional new particle formation events. It is misleading, since many of the anthropogenic particles also are formed in through nucleation processes, as the authors know. This should be revisited through the manuscript.

The word background seems to be applied with (at least) two different meanings, one for the background site and one for the background concentrations (e.g. lines 159-161, lines 226-227), which here, if I understood correct, refers to concentration of accumulation mode particles in general. Additionally, it seems that on lines 322-323 and 329 the term background means the concentration without nucleation event, otherwise "increment of background concentration on nucleation days" would mean higher N100-1000 than on non-nucleation days. Why not simply use the term accumulation mode?

[Figure]

Specific notices:

Lines 41-43, should these sentences be one?

Line 68-70: Open (NH4)2SO4, NH4NO3 and TC

Line 105: N6-100 referred to as Aitken mode, should be nucleation + Aitken mode, compare e.g. to lines 22-23.

The time over which the daily mean is calculated for seasonal or annual NSFs should be mentioned in the methodology part (perhaps line 130), now it appears only on lines 322-323.

Line 131: reference for the site in question, in some/many locations more nucleation days.

Lines 143-144: describe shortly the "second group"

Lines 159-160: What are the background concentrations meant here? In the context of the table it is logically connected to background site, but from the sentence it seems not to be so.

Line 277: maybe "of what" instead of "which", or modify the sentence otherwise.

Lines 384-387: It seems that the growth of these particles is considered as a loss of these particles in this analysis. The share of particles grown out of nucleation mode size range should be possible to determine from the dmps measurements with the normal methodologies (e.g. Kulmala et al., 2012, referred to in the manuscript).

Lines 404-405: I don't believe the authors mean the particles of e.g. 20 nm diameter are thermodynamically unstable.
* * *

---

## Author Comment (AC2) · 27 Nov 2017

**Response to Referee #2**

The authors thank Referee #2 for his/her valuable comments. We utilized all of them to further improve and clarify the MS, and made several extensions and alterations. Our responses to the comments are as follows.

**Major comment**

1. The actual meaning of the NSF is not clear, due to the normalization of nucleation + Aitken mode concentration with the accumulation mode (why not e.g. with the mean N6-100 from 6 to 9 am on the same morning?). The applied normalization may cause unintended signals: for example, if we consider two days during which the N6-100 is exactly similar, but on the latter (called here a nucleation day just to show the point) N100-1000 is lower than on the first by a factor of 1.5, the NSF would by 1.5. This would not only be a false signal but also to the wrong direction: during lower sink conditions the equal source should lead to higher concentrations, and if equal N6-100 was observed, the source should be weaker and thus NSF smaller than 1. This does not necessarily mean that the applied definition of NSF would not make sense, but it's behaviour with the applied data set should be analysed and its meaning explained much more in detail.

We can agree with the Referee that the applicability and exact meaning of the nucleation strength factors (NSFs) is complex despite their relatively simple mathematical definition. We extended the corresponding part of the text with several new aspects and made the existing explanations more explicit and clear to avoid any misunderstanding. We further emphasized the assumptions for their utilization and their rigorous interpretation, and also named the two versions of the NSF differently (as $NSF_{NUC}$ for the concentration increment on a nucleation day, and $NSF_{GEN}$ for the concentration increase on a general day) to assist their differentiation. The changes are highlighted in red. The conclusion of the Referee based mainly on two specific examples in this paragraph, however, cannot be accepted. A) The normalisation of $N_{6-100}$ cannot be performed e.g. to the mean $N_{6-100}$ from 06:00 to 09:00 on the same morning (as suggested by the Referee) because the effect of NPF and particle growth process can continue till the next morning for some events, and therefore, it can contribute to an elevated $N_{6-100}$ in mornings, which would disturb the correct quantification. This effect shows up as an elevated baseline in the morning in Fig. 1, and its consequences were further discussed in section 3.2. B) One of the basic assumptions for the NSFs is that the major emission and formation processes of the UF particles except for NPF are uniformly present on both nucleation and non-nucleation days

(lines 119–120 of the original MS). This can be fulfilled by taking into account concentration data for several days, and it can be misleading to deal with just two specific days, i.e. with one nucleation day and one non-nucleation day. More importantly, these days cannot be characterised by identical $N_{6–100}$ at all because this would seriously contradicts with the equality of all sources except for NPF (you simply cannot have identical $N_{6–100}$ on a non-nucleation day and on a nucleation days if the other sources - except for NPF - are equal). The basic assumption of the NSF is evidently not met for this specific example, and the final conclusion drawn by the Referee is then inaccurate. As far as the minimum number of days (more exactly, the minimum number of NPF events during a time interval) sufficient for obtaining representative NSF values is concerned, it was discussed in section 3.2 that 2–3 weeks in winter (which is the most unfavourable interval from this point of view, see Table 1) might not be fully satisfactory for this purpose. At the same time, the longer time interval needed does not detract from the value of the quantification, because the health and environmental effects of NPF are important mostly on longer time scales.

**Minor comments, on the terminology**

2. The word nucleation is used in the manuscript for regional new particle formation events. It is misleading, since many of the anthropogenic particles also are formed in through nucleation processes, as the authors know. This should be revisited through the manuscript.

The word nucleation is used in the MS to express the regional- and urban-type NPF. The particles generated by these processes are formed in the ambient air from precursor gases, and are of secondary character. The particles which are formed inside a localised source or within a plume are emitted directly into the air from their emission sources, and are regarded as primary particles. We follow this pragmatic concept throughout our publications.

3. The word background seems to be applied with (at least) two different meanings, one for the background site and one for the background concentrations (e.g. lines 159-161, lines 226-227), which here, if I understood correct, refers to concentration of accumulation mode particles in general. Additionally, it seems that on lines 322-323 and 329 the term background means the concentration without nucleation event, otherwise "increment of background concentration on nucleation days" would mean higher N100-1000 than on non-nucleation days. Why not simply use the term accumulation mode?

We can agree with the Referee, and modified the whole text at many places to distinguish between spatial (near-city) background and aerosol concentration background ($N_{100–1000}$). We worked with size intervals of 6–100 nm and 100–1000 nm, and mentioned it explicitly now

that the intervals estimate the nucleation + Aitken modes, and accumulation modes, respectively in most cases.

**Minor comments, specific notices**

4. Lines 41-43, should these sentences be one?

The two sentences were joined as requested.

5. Line 68-70: Open (NH4)2SO4, NH4NO3 and TC

The meaning of the first two chemical formulae is unambiguous (ammonium sulfate and ammonium nitrate, respectively). We explained the abbreviation of the TC now as total carbon contained in particles (TC, TC=EC+OC).

6. Line 105: N6-100 referred to as Aitken mode, should be nucleation + Aitken mode, compare e.g. to lines 22-23.

The request was adopted.

7. The time over which the daily mean is calculated for seasonal or annual NSFs should be mentioned in the methodology part (perhaps line 130), now it appears only on lines 322-323.

The request was adopted.

8. Line 131: reference for the site in question, in some/many locations more nucleation days.

An overview paper by Nieminen et al., Global analysis of continental boundary layer new particle formation based on long-term measurements to be submitted very soon was added as a reference for this general property.

9. Lines 143-144: describe shortly the "second group"

Main features of the referred group of the monthly mean nucleation frequency distribution were described now explicitly as: The seasonal variation of the nucleation frequency fits into the second group of the measurement sites - which is characterised by the highest number of nucleation events in spring and the lowest in winter, with relatively high total number of events (Manninen et al., 2010).

10. Lines 159-160: What are the background concentrations meant here? In the context of the table it is logically connected to background site, but from the sentence it seems not to be so.

The whole text was modified at many places to distinguish between spatial (near-city) background and concentration background ($N_{100–1000}$).

11. Line 277: maybe "of what" instead of "which", or modify the sentence otherwise.

The sentence was modified as requested.

12. Lines 384-387: It seems that the growth of these particles is considered as a loss of these particles in this analysis. The share of particles grown out of nucleation mode size range should be possible to determine from the dmps measurements with the normal methodologies (e.g. Kulmala et al., 2012, referred to in the manuscript).

The residence time of particles with diameters 6–25 nm determined in this way indeed includes the major sinks, namely the coagulation of particles and growth out of particles from the specified size range. Their relative contributions (mean coagulation rate and growth out rate with respect to the formation rate $J_6$) in Budapest are evaluated in an ongoing study, and they are to be reported and discussed in a separate MS.

13. Lines 404-405: I don't believe the authors mean the particles of e.g. 20 nm diameter are thermodynamically unstable.

The statement was indeed meant for the aerosol system containing particles with diameters below 20 nm, and not for the particles themselves. The sentence was corrected accordingly.

Finally, we would like also to mention that we considered several options and size intervals for quantifying the NPF as a single source of particles, and found that the quantities presented in the MS are the most advantageous and expressive possibilities. We are aware that the treatment introduced has some limitations. These are discussed explicitly and in detail in the revised paper. Nevertheless, the proposed method is capable of quantifying the relevance of particles from NPF relative to other sources, e.g. to road traffic emissions in cities for the first time, which is an unambiguous and important step forward in urban atmospheric studies.

Imre Salma
27 November 2017

---

## Author Comment (AC1)

**Response to Referee #1**

The authors thank Referee #1 for his/her valuable comments to further improve and clarify the MS. We have considered all recommendations, and made the appropriate alterations. Our specific responses to the comments are as follows.

1. Two NSF factor have been used in the manuscript. The mixed usage of the two factors always confused me. I would like to suggest the authors to clearly state which factor they are referring to whenever possible, e.g., NSFs in Line 134-136. Also, is NSFnucl days larger than NSFall days? Looks to me that (N6-100/N100-1000)nul days is larger than (N6-100/N100-1000)all days by definition, isn't it? I would like to see more discussion on the relationship between the two factors.

We named the two versions of the NSF differently as $NSF_{NUC}$ for the concentration increment on a nucleation day, and $NSF_{GEN}$ for the concentration increase on a general day all over the MS to assist their differentiation. The interpretation of their meaning regarding the limiting values is valid for both of them. We also extended the text with several new aspects on their relationships. The changes are highlighted in red.

2. In table 2, wouldn't I get (N6-100/N100-1000)nul days from the numbers listed? For example, (N6-100/N100-1000)nul days can be obtained (1.72/1.03 = ((N6-100)Nucl/(N6-100)Non-nucl)/((N100-1000)Nucl/(N100-1000)Non-nucl) = ((N6-100)Nucl/(N100-1000)Nucl)/((N6-100)Non-nucl/(N100-1000)Non-nucl) = (N6-100/N100-1000)nul days. However, my number is different from those listed in Table 3. What is the problem?

In data processing, concentration ratios were first derived from the individual data; later these ratios were averaged for nucleation and non-nucleation days for a specific time point considered, and finally the NSF values were calculated and averaged over the time interval selected. This does not necessarily and exactly results in the same final NSF value as that derived from the mean concentrations. The former data treatment is the correct method considering the dynamic character of atmospheric concentrations, uncertainties of the data, and the propagation of errors. This was actually utilised in the MS.

3. (Line 156-158), put "Data coverage for summer and autumn for the mean ratios" as a footnote of Table 2.

The requested footnote was added.

4. (Line 159-160) the lower background particle concentration on nucleation days in winter came from real measurements, right?

All the primary data presented were obtained experimentally, which was confirmed now in this section as well.

5. (Line 173) rephrase "its consideration in the averaging is justified".

The sentence was modified to clarify its meaning.

6. Clearly mark the vertical coordinates (NSFnucl days or NSFall days) in Figures 1 & 4, and also include NSFnucl days or NSFall days in the body instead of the title of Tables 3 & 4.

The requested changes were fully adopted. See also the response to Comment 1.

7. In table 2, what caused the behavior of N6-100 on NPF days? Especially, the peak at night. Also, please expand to discuss why N6-100 is significantly larger between 6-9 am on non-nucleation days? Does this mean that nucleation was hindered by the high concentrations of preexisting particles?

We assume that the Referee meant Figure 2 instead of Table 2. The late evening/night peak was observed from the beginning of the measurements in Budapest (see e.g. Salma et al., 2010), and can likely be related to the combined effect of burning and heating activities at residences and homes, and of local meteorology. They are also influenced by the daily cycling of the boundary layer mixing height and mixing intensity. The exact interpretation of the evening peak, however, needs further dedicated investigations. As far as the $N_{6-100}$ between 6 and 9 am is concerned, its higher level on non-nucleation days with respect to nucleation days is indeed related to higher pre-existing aerosol concentration level, and thus, larges condensation sink values, which hinder the chances for NPF. These explanations were now included into the text.

8. (Line 272) lower solar radiation in winter is understandable, but less biogenic pre-cursor gases are not justified. Is there any evidence that NPF in Budapest requires biogenic vapors?

There is indirect evidence that biogenic emissions contribute to the early stage of the growth process, and likely to the nucleation itself as well. Distribution of the monthly mean occurrence frequency, and relatively low (<20%) relative contribution of gas-phase $H_2SO_4$ to the growth rate of particles can indicate the role of biogenic precursors. The sentence was extended to include this information, and two specific references were given now for further details.

9. (Line 277-279) rephrase "this observation raises the question ... for calculating the NSF".

The sentence was modified.

10. Regarding the health effects (Line 385-387), I would like to suggest the authors to be more conservative. The relative short lifetime is one aspect, but the toxicity per particle is another aspect. The authors just can't evaluate the health effects of nanoparticles generated by NPF.

We can fully agree. The discussion on the health effects was modified to be more accurate, and the specific aspect raised by the Referee was included.

Imre Salma
27 November 2017